# Improving Generation of Cardiac Organoids from Human Pluripotent Stem Cells Using the Aurora Kinase Inhibitor ZM447439

**DOI:** 10.3390/biomedicines9121952

**Published:** 2021-12-20

**Authors:** Su-Jin Lee, Hyeon-A Kim, Sung-Joon Kim, Hyang-Ae Lee

**Affiliations:** 1Department of Predictive Toxicology, Korea Institute of Toxicology (KIT), Daejeon 34114, Korea; sujin.lee@kitox.re.kr (S.-J.L.); hyeona.kim@kitox.re.kr (H.-A.K.); 2Department of Physiology, Seoul National University College of Medicine, Seoul 03080, Korea

**Keywords:** human cardiac organoid, aurora kinase, cell differentiation, ZM447439

## Abstract

Drug-induced cardiotoxicity reduces the success rates of drug development. Thus, the limitations of current evaluation methods must be addressed. Human cardiac organoids (hCOs) derived from induced pluripotent stem cells (hiPSCs) are useful as an advanced drug-testing model; they demonstrate similar electrophysiological functionality and drug reactivity as the heart. How-ever, similar to other organoid models, they have immature characteristics compared to adult hearts, and exhibit batch-to-batch variation. As the cell cycle is important for the mesodermal differentiation of stem cells, we examined the effect of ZM447439, an aurora kinase inhibitor that regulates the cell cycle, on cardiogenic differentiation. We determined the optimal concentration and timing of ZM447439 for the differentiation of hCOs from hiPSCs and developed a novel protocol for efficiently and reproducibly generating beating hCOs with improved electrophysiological functionality, contractility, and yield. We validated their maturity through electro-physiological- and image-based functional assays and gene profiling with next-generation sequencing, and then applied these cells to multi-electrode array platforms to monitor the cardio-toxicity of drugs related to cardiac arrhythmia; the results confirmed the drug reactivity of hCOs. These findings may enable determination of the regulatory mechanism of cell cycles underlying the generation of iPSC-derived hCOs, providing a valuable drug testing platform.

## 1. Introduction

Fifty-four percent of all drugs developed between 1998 and 2008 were eliminated at the clinical stage of development and thereby not approved [1,2]. Candidate drugs are generally eliminated at this stage because of their low efficacy or safety issues, including cardiac toxicity [3]. Therefore, the results of cardiac toxicity and efficacy evaluations play a major role in determining the success or failure of drugs that are under development [4]. Existing experimental methods used for cardiac toxicity and efficacy evaluations at non-clinical stages include two-dimensional culture cell models and in vivo animal models; the former exhibit advantages of having a large volume of available research data and are easier to use in experiments compared to in vivo models [5]. However, because in vivo environments cannot be simulated in cell models, it is difficult to predict clinical-stage results using these data [6]. Although in vivo animal models can be used to simulate in vivo environments, they are costly and involve time-consuming procedures [7]. Drug reactivity can also vary between humans and animal models [8]. Moreover, it tends to be reduced in animal studies due to ethical problems [9,10]. A human cardiac organoid (hCO) model derived from human-induced pluripotent stem cells (hiPSCs) has been proposed for overcoming the limitations of existing models [11,12,13,14]. Compared to iPSC-derived cardiomyocytes (iPSC-CMs), hCOs are similar to the human heart, as they include cardiac fibroblasts, cardiac endothermic cells, and various other cells that make up the actual heart [12,15]. For use as a drug evaluation model, iPSC-hCOs must be produced using a specific differentiation method and show similar drug evaluation results as the adult heart [16]. Compared to cell line and culture methods, studies using iPSC-hCOs have shown heterogeneous differentiation results. Various causes of differentiation have been identified, with cell confluence shown to affect differentiation [17]. The differentiation mechanism also varies according to the cell cycle ratios of cells due to the S/G2/M and G1 ratios are high when the cell cycle ratio is 50–70% [18].

Aurora kinase is the main kinase involved in the G2/M phase of the cell cycle by controlling kinetic progression [19]. Aurora kinase consists of three members (Aurora A, B, and C), among which Aurora A (Aur-A) is associated with centrosome maturation, separation, spindle formation, and chromosome separation [20]. Knockdown of Aur-A in embryonic stem cells (ESCs) increases the expression of p53, which is known to affect the expression of ectodermal and mesodermal genes [21]. In addition, in ESCs, the p53 family has been demonstrated to regulate mesoderm specification by adjusting the expression of *Wnt3* [22]. Therefore, we hypothesized that alterations in the expression of Aur-A during the differentiation process may could affect the cell composition of iPSC-hCOs and alter CM differentiation. To explore this hypothesis, we used ZM447439 (ZM), an Aur-A inhibitor, in several differentiation stages when producing and analyzing hCOs. Furthermore, we treated the produced hCOs with major ion channel drug inhibitors to confirm their reactivity to specific channels. We aimed to determine whether the production efficiency of iPSC-hCOs can be increased using cell cycle-regulating drugs and explore whether iPSC-hCOs can be used as a drug-testing model in non-clinical studies.

## 2. Materials and Methods

### 2.1. Cell Line and Culture Protocol

hFSiPS1 cells, which are hiPSCs from dermal fibroblasts, were obtained from the National Stem Cell Bank of Korea (Korea National Institute of Health). The hiPSCs were expanded and maintained in 6-well plates coated with human embryonic stem cell (ESC)-qualified Matrigel (Corning Inc., Corning, NY, USA). The hiPSCs were cultured in mTeSR™1 medium (Stemcell Technologies, Vancouver, DC, Canada) with daily media changes and passaged twice per week using Versene solution (Gibco, Grand Island, NY, USA). The cells were resuspended mTeSR™1 medium containing 10 μM Y27632 (Stemgent, Cambridge, MA, USA) on the first day after passing in a 37 °C, 5% CO_2_ incubator.

### 2.2. Quality Control of hiPSCs

#### 2.2.1. Flow Cytometry

The hiPSCs were dissociated with Versene solution and strained using a 40 µm nylon cell strainer. The cells were resuspended in cell sorting buffer containing Hanks Balanced Salt Solution (HBSS, Gibco), 4% Fetal Bovine Serum (FBS, HyClone, South Logan, UT, USA), 10 mM Hydroxyethyl Piperazine Ethane Sulfonic acid (HEPES, Gibco), 1X penicillin-streptomycin (Gibco), and 10 µM Y27632. hiPSCs were sorted using the human Induced Pluripotent Stem Cell Analysis and Sorting Kit (BD Biosciences, San Jose, CA, USA) according to the manufacturer’s instructions. Briefly, hiPSCs were incubated with TRA-1-60-PE, CD13 PerCP-Cy5.5, and SSEA-4 Alexa Fluor 647 (1:100) on ice for 20 min. After washing, the cells were evaluated on a FACS Aria^TM^ Fusion Flow Cytometer (BD Biosciences). The sorted cells were cultured in 6-well plates coated with Matrigel.

#### 2.2.2. PSC Scorecard Analysis

To confirm the pluripotency of the sorted hiPSCs, we performed real-time polymerase chain reaction (PCR) with a commercialized human pluripotent stem cell (hPSC) scorecard panel, including a combination of differentiation lineage markers and self-renewal, control, and housekeeping genes. cDNA was synthesized from 1 μg total RNA using a High-capacity cDNA Reverse Transcription Kit (Applied Biosystems, Foster City, CA, USA), and quantitative real-time PCR (qRT-PCR) was performed using 2X TaqMan Gene Expression Master Mix according to the TaqMan^®^ hPSC Scorecard™ Kit 96w fast assays (Life Technologies, Carlsbad, CA, USA) on a StepOne™ Real-Time PCR System (Applied Biosystems) according to the manufacturer’s instructions. Data were analyzed using TaqMan hPSC Scorecard analysis software (https://apps.thermofisher.com/hPSCscorecard/home.htm, accessed on 3 June 2020).

### 2.3. Differentiation of hiPSCs into CMs

The protocol for cardiomyocyte differentiation was modified from a protocol described previously [23,24]. Briefly, hiPSCs were seeded into plates (1.82 × 10^4^ cells/cm^2^) coated with hESC-qualified Matrigel on the day before differentiation. The media were changed daily and, after 2 days, with the media containing the Matrigel. On day 0, differentiation was initiated at 80–90% confluency with differentiation basal medium (RPMI-1640 supplement minus insulin B27, Gibco) and 10 μM CHIR99021 (Selleck Chemicals, Houston, TX, USA). After 24 h, CHIR99021 was removed. On day 3, the medium was replaced with fresh differentiation basal medium containing 5 μM IWP4 (Stemgent) for 2 days. The medium was changed to differentiation basal medium from days 5 to 6. The maintenance medium (RPMI-1640 supplement B27, Gibco) was refreshed every 2 days. ZM was purchased from Toronto Research Chemicals (North York, ON, Canada) and prepared as a stock solution in dimethyl sulfoxide (DMSO).

### 2.4. Differentiation of hiPSCs into hCOs

hCO differentiation was performed as described previously with slight modifications of the differentiation basal medium [25]. The hiPSCs were seeded at 1 × 10^6^ cells/well in ultra-low attached 6-well plates in mTeSR™1 medium containing 5 µM Y27632 for 1 day. The cells were incubated in a 37 °C, 5% CO_2_ incubator to allow embryonic body formation (EB). As the differentiation basal medium, RPMI-1640/B27 medium (Gibco) without insulin and containing 0.2 mg/mL human L-ascorbic acid (Sigma-Aldrich, St. Louis, MO, USA) was used. On day 0, the medium was replaced with basal medium containing 10 μM CHIR99021 (Stemgent). After 48 h, the medium was replaced basal medium with 5 μM IWP4 (Stemgent) for 2 days. From days 4 to 7, the medium was changed to differentiation basal media, and the maintenance medium was changed every 2–3 days.

### 2.5. Cell Cycle Analysis

The hiPSCs were dissociated using TrypLE^TM^ express (Gibco) for 4 min. The cells were resuspended in 400 μL cold PBS (HyClone) and fixed with 800 μL 100% ethyl alcohol (Sigma-Aldrich) overnight at 4 °C, and then incubated with propidium iodide (PI, Life Technologies) and RNase (Invitrogen, Carlsbad, CA, USA) for 30 min. Data were acquired using a Flow Cytometer in CytoFLEX (Beckman Coulter, Brea, CA, USA).

### 2.6. RNA extraction, cRNA synthesis, and qRT-PCR

Total RNA was isolated from hiPSCs, hiPSC-CMs, and hCOs using TRIZol (Life Technologies) at 4 weeks. Total RNA was used for first-strand cDNA synthesis by using a GoScript Reverse Transcription Mix kit (Promega, Madison, WI, USA). qRT-PCR was carried out using GoTaq qPCR Master Mix (Promega) according to the manufacturer’s instructions. qRT-PCR was performed on StepOne™ Real-Time PCR System. The data were analyzed using the ΔCt methods and normalized to the GAPDH gene expression [26]. The primer sequences were in Table 1.

### 2.7. Immunofluorescent Staining

The cells were fixed with 4% paraformaldehyde (Biosesang, Gyeonggi-do, Korea) for 20 min at room temperature and permeabilized with 0.5% Triton-X-100 (Sigma-Aldrich) in PBS for 30 min. The cells were blocked with 5% bovine serum albumin (BSA) (Rocky Mountain Biologicals, Missoula, MT, USA) for 20 min and incubated with primary antibodies against cTnT (Abcam, Cambridge, UK), CD31 (R&D Systems, Minneapolis, MN, USA), and α-SMA (Sigma-Aldrich) at a 1:100 dilution at 4 °C overnight. Secondary antibodies conjugated with fluorescent dyes (Thermo Fisher Scientific, Waltham, MA, USA) were stained with 1:200 dilutions overnight at 4 °C and mounted Mounting Medium with DAPI-Aqueous (Abcam). Imaging was performed by confocal microscopy on an Olympus FV3000 (Tokyo, Japan). All confocal images were processed with FV31S-SW.

### 2.8. Electrophysiology

The ion channel currents of iPSC-CMs were measured in singles cells for study. The cells were transferred onto Matrigel-coated glass coverslips in 4-well culture plates and maintained in a culture incubator at 37 °C and 5% CO_2_. Currents and voltages were recorded in whole-cell patch-clamp recordings (pClamp 10.2, Digidata 1440, Molecular Devices, Sunnyvale, CA, USA) performed at 36.5 °C in an external solution containing 145 mM NaCl, 5.4 mM KCl, 10 mM HEPES, 1.8 mM CaCl_2_, 1 mM MgCl_2_, and glucose 5 mM and internal solution containing NaCl 5 mM, 6 mM ethylene glycol *bis*(2-aminoethylether)-*N*,*N*,*N*′,*N*′-tetra acetic acid (EGTA), 10 mM HEPES, 5 mM Mg-ATP, 20 mM KCl, 120 mM K-Asp, and 2 mM CaCl_2_. The pH was adjusted 7.4 with 3 M NaOH solution. In voltage-clamp mode, a standardized step protocol was used to elicit the major cardiac *I_Ca_* (calcium channel currents) and *I*_hERG_ (*hERG* channel currents). These currents were verified using the specific blockers nifedipine for *I*_Ca_ and E-4031 for *I*_hERG_, respectively. All reagents for solution formulation were purchased from Sigma-Aldrich.

### 2.9. Transcriptome Analysis

RNA (1 µg) was used as input material for RNA sample preparations. Sequencing libraries were generated using NEBNext^®^ Ultra TM RNA Library Prep Kit for Illumina^®^ (New England Biolabs, Ipswich, MA, USA) following the manufacturer’s recommendations, and index codes were added to attribute sequences to each sample. Briefly, mRNA was purified from total RNA using poly-T oligo-attached magnetic beads. Fragmentation was carried out using divalent cations under elevated temperature in NEBNext First Strand Synthesis Reaction Buffer (5X). First-strand cDNA was synthesized using random hexamer primer and M-MuLV Reverse Transcriptase (RNase H-). Second strand cDNA synthesis was performed using DNA Polymerase I and RNase H. Remaining overhangs were converted into blunt ends via exonuclease/polymerase activities. After adenylation of the 3′ ends of the DNA fragments, an NEBNext Adaptor with a hairpin loop structure was ligated for hybridization. To select cDNA fragments of preferentially 150–200 bp in length, the library fragments were purified with the AMPure XP system (Beckman Coulter). Next, 3 µL USER Enzyme (New England Biolabs) was used with size-selected, adaptor-ligated cDNA at 37 °C for 15 min followed by 5 min at 95 °C before PCR. PCR was performed using Phusion High-Fidelity DNA polymerase, Universal PCR primers, and Index (X) Primer. The PCR products were purified (AMPure XP system), and the library quality was assessed on an Agilent Bioanalyzer 2100 system (Agilent Technologies, Santa Clara, CA, USA). The c index-coded samples were clustered on a cBot Cluster Generation System using PE Cluster Kit cBot-HS (Illumina, San Diego, CA, USA) according to the manufacturer’s instructions. The library preparations were sequenced on an Illumina platform, and paired-end reads were generated. To analyze the functions of genes showing increased expression, we conducted enrichment analysis using the Gene Ontology (GO), Kyoto Encyclopedia of Genes and Genomes (KEGG), and Reactome databases; these databases identify genes according to their biological processes and pathways.

### 2.10. Multi-Electrode Array Assay

The multi-electrode array (MEA) plate containing electrodes was coated with Matrigel before transfer to iPSC-hCOs. The hCOs were seeded onto the MEA plate and allowed to attach to wells for 30 min at 37 °C. A full change medium on the plates was changed at least 4 times before the recordings. Field potentials (FP) were recorded from spontaneously beating iPSC-hCOs. During the recording period, the MEA plate was placed at 37 °C in a sterile environment and used the Maestro MEA system (Axion Biosystems, Atlanta, GA, USA). Data were filtered with a Butterworth 0.1 Hz–2 kHz band-pass filter. The hCOs were sequentially exposed to increasing concentrations of test drugs. Drug treatment was repeated at 20 min. The FP waveforms were recorded for 5 min, and the last 1 min was used for data analysis. Fridericia’s formula (FPD_cF_), where FPD_cF_ = FPD/Beat Period^0.33^), was used to correct the field potential duration (FPD) dependence on the beating rate.

### 2.11. Statistical Analysis

All experimental data were analyzed using GraphPad Prism (GraphPad, San Diego, CA, USA), pCLAMP (Axon Instruments, Foster City, CA, USA), Origin 8 (OriginLab Corp, Northampton, MA, USA), and Excel (Microsoft, Redmond, WA, USA) software for statics and graph production. All values are presented as the mean ± S.E.M., and N equals the number of the data. Statistical significance was determined using the Student’s t-test and one-way ANOVA with post hoc testing using Dunnett’s method; *p* < 0.05 was considered to indicate statistical significance

## 3. Results

### 3.1. Differentiation of CMs from Quality Controlled hiPSCs

We determined the efficiency of differentiation using hiPSCs based on their pluripotency. To enhance pluripotency, we performed cell sorting to generate hCOs from hiPSCs. Previous studies showed that the fibroblast marker CD13 is downregulated in hiPSCs, whereas the pluripotency markers stage-specific embryonic antigen 4 (SSEA-4) and TRA-1–60 are upregulated [27]. Here, fluorescence-activated cell sorting (FACS) revealed the negative selection for CD13 (98.84%) of hiPSCs (Figure 1A, left). Among the CD13-negative population, we detected a double-positive result for the SSEA-4/TRA-1-60 population (61.53%, Figure 1A, right). Scorecard analysis showed that the expression of pluripotency markers was significantly upregulated, and that lineage-specific markers (mesendoderm, ectoderm, mesoderm, and endoderm) were downregulated in hiPSCs (Figure 1B), confirming that pluripotency was achieved.

### 3.2. ZM Treatment Changes Mesoderm Subtype during Differentiaion of iPSC-CMs 

Previous studies have shown that during CM differentiation in hiPSCs, increases in cell confluence increase the G1 ratio and decrease the G2/M/S ratio [18]. Therefore, we hypothesized that treatment with ZM before differentiation would increase the G1 ratio by inhibiting Aur-A, thereby increasing the differentiation efficiency of CMs. Furthermore, as the expression of Aur-A is known to be associated with the mesoderm and Wnt/Gsk3 in hiPSCs, we predicted that treatment with ZM during differentiation would also affect the differentiation of CMs [21,22,28]. To confirm this, we performed ZM treatments before and after the differentiation stages on days -4 and -1, as well as before and after the mesoderm stage in the CM differentiation stage (Figure 2A). To determine the appropriate treatment concentration of ZM, we applied treatments ranging from 0 to 100 nM for 5 days in hiPSCs. The results revealed that cell viability was not affected at a minimum concentration of 10 nM and maximum concentration of 100 nM, and thus these values were adopted as limits for further analysis (data not shown). We analyzed changes in the cell cycle stage under the selected conditions of cells treated with ZM for 24 h (Figure 2B). The G1 ratio was lowest on D-4, as the degrees of cell confluence at other differentiation stages were higher than that shown in Figure 2C. To investigate changes in S/G2/M phase following ZM treatment, we compared a ZM-treated sample (fold-changes) to a control sample that was not treated with ZM. Following treatment with ZM on D1, we found that cells in G1 phase were significantly increased by 100 nM ZM treatment (Figure 2C, middle); however, S/G2/M phase cells were significantly decreased (Figure 2C, down). Furthermore, a cell cycle change occurred at the CM differentiation stage following ZM treatment.

Analysis of the mRNA expression in iPSC-CMs at 30 days after differentiation confirmed that the mesoderm subpopulation of CMs was differentiated by ZM treatment. In addition to the myocardial cell marker cardiac troponin T (*cTnT*), we confirmed the expression of the fibroblast (FB) marker vimentin and endothermic cell (EC) marker vascular endothelial cadherin (*VE-Cad*), which belongs to the mesoderm. The expression of *cTnT* increased significantly following ZM treatment on D-4, D1, and D3, whereas that of vimentin increased significantly on D-1, at 10 nM but decreased significantly at 100 nM. *VE-Cad* expression increased significantly on D-1. On D3, the gene expression of *Vimentin* and *VE-cad* was higher than that in the control following ZM treatment. Thus, addition of ZM during CM differentiation affects the mesoderm subpopulation, particularly CM differentiation (Figure 3A). The main functions of CMs are related to electrophysiological properties; they are associated with the expression of major cardiac ion channels (*Cav1.2, Nav1.5*, and human ether-a-go-go-related gene (*hERG*)). To examine the functional changes in CMs caused by ZM, we examined their mRNA expression, which showed that the expression of *Cav1.2, Nav1.2*, and *hERG* increased significantly during ZM treatment on D-1 and D3 (Figure 3B). Therefore, we confirmed that treatment with ZM during the differentiation of CMs also affects the expressions of ion channels related to CM functions.

### 3.3. Improvement in Cardiac Gene Expression in ZM-Treated iPSC-hCOs

Cardiac organoids produced by self-organizing methods consist of a variety of cell types, and exhibit batch-to-batch variations [17]. Therefore, we treated cardiac organoids with ZM to confirm whether there was a significant change in the mesoderm subpopulation. Specifically, we applied ZM treatments before hCO differentiation and after the mesoderm step by referring to the iPSC-CM differentiation process (Figure 4A,B). After 30 days of differentiation, we confirmed mRNA expression using CM, FB, and EC markers to determine the cell composition of the produced hCOs. We found that CM, FB, and EC markers showed significant increases following ZM treatment on D2. Additionally, treatment with ZM on D-1 and at the cardiac mesoderm stage D4 (before differentiation) also significantly reduced the mesoderm subpopulation (Figure 4C). Immunofluorescence confirmed the expressions of cells belonging to the mesoderm, and EC was confirmed by the marker expression for CD31. We also confirmed that CM and FB were expressed in hCOs produced using antibodies for cTnT and α-smooth muscle actin (α-SMA) (Figure 4D). Therefore, treatment with ZM during the differentiation process of hCOs affected the gene expression of cells belonging to the mesoderm. Particularly, treatment with ZM after the mesoderm stage increased the expressions of CM, FB, and EC markers, reflecting mesoderm subpopulations.

### 3.4. Increase in Cardiac Ion Channel Function in ZM-Treated iPSC- hCOs

To determine whether treatment with ZM affects cardiac function, we examined the mRNA expression of major cardiac ion channels in ZM-treated iPSC-hCOs. During ZM treatment on D-1, the mRNA expressions of *Cav1.2* and *Nav1.5* increased significantly, whereas on D2, *Cav1.2* (calcium channel gene), *Nav1.5* (sodium channel gene)*,* and *hERG* expression all increased significantly (Figure 5A). We measured the peak current amplitude of each ion channel using the whole-cell patch-clamp technique, which is an electrophysiological method. These currents were verified using specific blockers. The *I*_Ca_ was recorded as the nifedipine-sensitive currents and *I*_hERG_ was recorded as the E-4031-sensitive currents. The *I*_Ca_ density increased twofold following ZM treatment (Figure 5B), whereas the *I*_hERG_ density tended to increase during ZM treatment, but not significantly (Figure 5C).

### 3.5. Increase in Cardiac Function-Related Gene Expression in ZM-Treated hCOs

We conducted transcriptome analysis to analyze the overall gene changes in hCOs differentiated by treatment with ZM (Figure 6). For the mRNA expression results of these hCOs, we compared mesoderm markers in the control, D2, and ZM 100 nM groups; we selected markers showing significantly increased ion channel expression for comparison with the control. We found 33,111 genes that were significantly expressed in both groups, with 140 and 11 showing increased and decreased expression, respectively, in the ZM-treated group (Figure 6A). The top five results for the three categories covered by GO categories (cellular component, molecular function, and biological process) were the contractile fiber part (GO:0044449), sarcomere (GO:0030017), contractile fiber (GO:0043292), myofibril (GO:0030016), and I band (GO:0031674) (Figure 6B). We used KEGG enrichment analysis to identify the pathways to which the significantly increased genes after ZM treatment belonged. The top five pathways were the complement and coagulation cascades (hsa04610), dilated cardiomyopathy (hsa05414), hypertrophic cardiomyopathy (hsa05410), bile secretion (hsa04976), and adrenergic signaling in CMs (hsa04261) (Figure 6C). The Reactome database is related to variable reactions and biological pathways in the human model. The top five results for this database were striated muscle contraction, intrinsic pathway of fibrin clot formation, muscle contraction, platelet degranulation, and response to elevated platelet cytosolic Ca^2+^ pathway (Figure 6D). The results described above confirm that genes related to cardiac function and maturity increased significantly during ZM treatment in the hCO differentiation stage.

### 3.6. Drug Responses of FPs in hCOs

We confirmed the expression of major ion channels by analyzing the molecular and electrophysiological properties of the produced hCOs. To enable use as a drug-responsive model for new drugs, it is necessary to verify that the main ion channels expressed perform normal drug reactions [29]. Therefore, we investigated the electrophysiological reactions to drugs using treatments with nifedipine, tetrodotoxin (TTX), and E-4031, which inhibit major ionic channels. We employed a MEA system, which is a non-invasive method; hCOs were attached to the MEA plate, which can read extracellular electric signals. We then sequentially applied treatments with ion channel inhibitors at low to high concentrations every 20 min.

Treatment with nifedipine, a Ca^2+^ channel blocker, significantly increased the beats per minute (BPM) of the hCOs to above 0.3 μM and significantly decreased their FPD_cF_. Treatment with TTX, an Na^+^ blocker, significantly reduced the FPA, and FP stop occurred in all batches at concentrations above 1 μM. The hERG blocker E-4031 caused a significant increase in FPD_cF_ above 0.003 μM, and samples exhibited arrhythmic occurrences. Therefore, the drug reactivity of major ion channels was significant in the produced hCOs (Figure 7B).

## 4. Discussion

HCOs are known to have a heart-like structure and cell composition [17]. They also have the advantage of being able to mimic the heart environment, allowing for accurate predictions of drug efficacy and safety evaluations, compared to conventional models [30]. There are two methods for producing hCOs using hiPSCs: (1) a direct method of differentiating each cell constituting the heart into two-dimensions (2D), followed by its conversion into a three-dimensional structure for co-culture [31,32,33,34,35,36,37,38,39]; and (2) a self-organizing method of creating a heart organ-like structure by aggregating stem cells [11,12,13,40]. Here, we selected the self-organizing method, in which various cell compositions were simultaneously constructed during the differentiation process. This approach can simulate cardiac physiological functions. However, the self-organizing method, it also suffers from several disadvantages. CMs consisting of self-organizing hCOs are relatively immature compared to CMs prepared by the direct method and they can exhibit cell-to-cell variation [41].

To overcome these limitations, we explored the possibility of using ZM, an inhibitor of Aur-A (which controls the cell cycle), in the differentiation stage. The differentiation of CMs is related to the cell cycle, and the Wnt/Gsk3β mechanism in CMs varies depending on the confluence of hiPSCs at the beginning of differentiation [18,42]. Our experimental results confirmed that the cell cycle ratio changed according to the confluence of hiPSCs. Furthermore, the cell cycle ratio changed following treatment with ZM.

Recently, studies reported the effects of Aur-A on differentiation mechanisms in stem cells [21,22]. Aur-A has been shown to support p53 activation in ESCs, and the inhibition of Aur-A in ESCs has been found to increase p53 activity and mesendoderm gene expression [21]. Furthermore, the activity of p53 family in ESCs influences Wnt signaling, and the p53 family is necessary for differentiation into the mesendoderm; it is affected by Wnt signaling during differentiation [22]. Overexpression of p53 in hiPSCs leads to differentiation into CMs, which results from p53 increasing the glycogen synthase kinase (GSK) initiation effect in the Wnt signaling pathway [43]. Following treatment with an inhibitor of the cyclin-dependent kinases (CDKs) that regulate the cell cycle in hiPSCs, the expressions of mesoderm markers decreased [44]. Therefore, we hypothesized that inhibiting the cell cycle controller Aur-A in hiPSCs would affect Wnt signaling via p53 and that this would affect differentiation of the mesoderm subpopulation, which comprises the main cellular composition of the heart [45]. Our results confirmed that treatment with ZM, an Aur-A inhibitor, during the differentiation process from iPSC to hCOs, increased not only CMs, but also other mesoderm subpopulations. Particularly, we observed a significant increase in mesoderm gene expression during simultaneous processing with a GSK inhibitor before the mesoderm stage. Further experiments are needed to determine whether ZM caused this phenomenon by increasing the efficiency of GSK inhibitors. In addition, ZM is known to inhibit not only Aur-A, but also Aur-B. Thus, it is also necessary to examine the differentiation process of hCOs using other Aur-A inhibitors [46].

We found that treatment with ZM during the hCO differentiation process influenced not only mesoderm marker expression, but also the expressions of major genes involved in the electrophysiological function of the heart. Furthermore, mRNA analysis revealed increased expression of *Cav1.2*, which is associated with Ca^2+^ handling, increased. Moreover, myofibril maturation (contractile fiber part, I band, Z disc), adrenergic signaling, and cardiac muscle contraction, which are associated with gene expression in mature CMs, were increased significantly following ZM treatment [47,48,49,50,51,52]. When we applied ZM treatment and measured the ion channel current in iPSC-CMs after differentiation, we found that the *I*_Ca_ density increased to compared with the control and the *I*_hERG_ tended to increase. Therefore, in the hCO differentiation process, treatment with ZM affects the structure and function of Ca^2+^ and CMs to induce maturation.

By measuring drug reactivity in the produced hCOs using MEA, we confirmed that our hCO model can function as a drug efficacy and safety evaluation model. In iPSC-CMs, nifedipine has been shown to selectively inhibit Ca^2+^ channels, increase the BPM, and decrease FPD_cF_; these phenomena were also reproduced in our experiments [53]. TTX, which inhibits the Na^+^ channel, exhibits cell-to-cell variations in iPSC-CMs; it has also been shown to primarily reduce amplitude [54]. We found that FPA also decreased in our model. E-4031, an hERG blocker, has been shown to maintain the beating rate in vivo, increase FPD in iPSC-CMs, and cause arrhythmia above 0.03 mM [55,56]. In experiments conducted using our developed hCO model, we found that that treatment with E-4031 significantly increased FPD_cF_, and that arrhythmia occurred at high concentrations. The in vitro cardiotoxicity testing models based on human stem cells could provide more relevant models for drug safety assessment. These models have the potentials to be used in non-clinical research in keeping with the 3Rs (refine, reduce, and replace) in animal testing and are more reliable than previous in vitro/in vivo assays.

## 5. Conclusions

Here, we proposed a new differentiation method to overcome the disadvantages of hCO production through the self-organizing method. We found that treatment with ZM, which inhibits the cell cycle controller Aur-A, increased the expression of non-myocytes and CMs during hCO induction. Furthermore, this treatment increased the expression of maturation-related genes in CMs. Our model can be used as a new drug-testing platform due to its reactivity to major cardiac ion channel blockers.

## Figures and Tables

**Figure 1 biomedicines-09-01952-f001:**
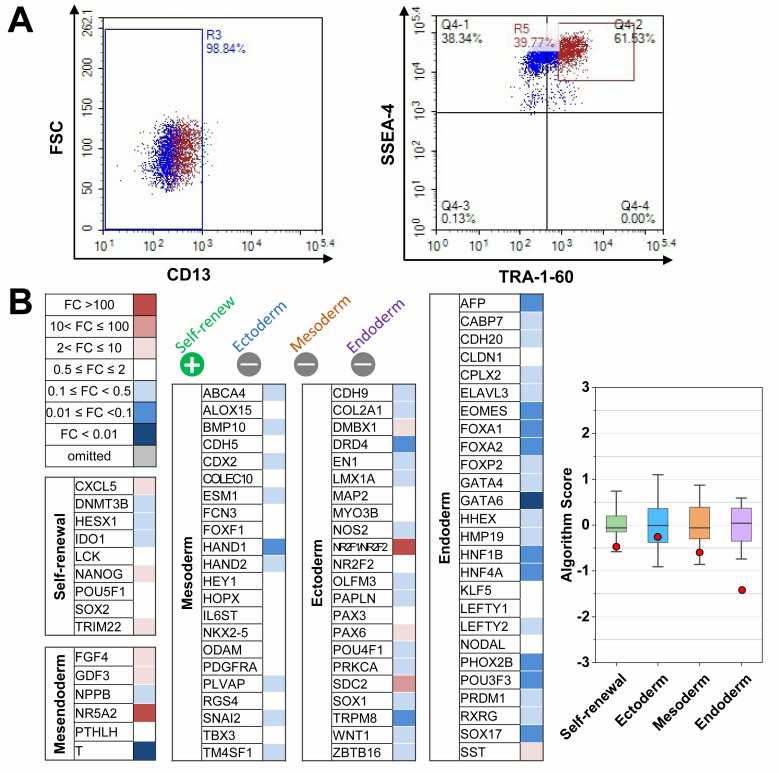
Quality control of human induced pluripotent stem cells (hiPSCs) hFSiPS1: (**A**) Representative FACS traces for CD13 (human fibroblast marker), TRA-1-60, and SSEA-4 (pluripotency markers) combination. Gating of CD13-negative population (left) and TRA-1-60-positive/SSEA-4 positive (right). (**B**) Scorecard analysis of sorted hiPSCs showing heatmaps of self-renewal genes and lineage-specific genes for mesendoderm, ectoderm, mesoderm, and endoderm (left). The Box-plot data from scorecard analysis (right). Red dots indicate sorted algorithm score of hiPSCs. Color box score is the range of scores of the undifferentiated reference stem cells set.

**Figure 2 biomedicines-09-01952-f002:**
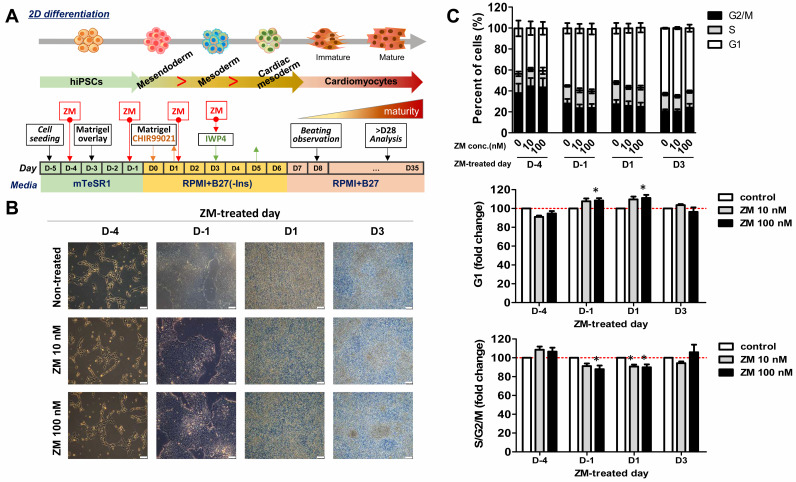
Effects of ZM447439 on the cell cycles of differentiated cardiac cells from hiPSCs: (**A**) Protocol of cardiomyocyte differentiation treated with ZM447439. (**B**) Differentiation of ZM-treated cardiomyocytes. Scale bar, 100 μm. (**C**) Cell cycle profile of cardiac-differentiated cells with 0, 10, and 100 nM ZM treatment (*n* = 3–4, top). G1 (middle) and S/G2/M proportions (down) of ZM-treated samples at D-4, D-1, D1, and D3, respectively. Data are presented as the mean ± SEM. * *p* < 0.05, by Dunnett’s test.

**Figure 3 biomedicines-09-01952-f003:**
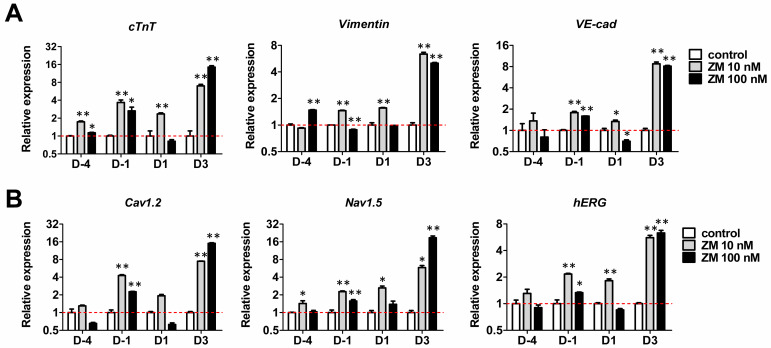
Effects of ZM447439 on cardiac gene expression of hiPSC-CMs: (**A**) Gene expression of the cardiomyocytes marker (*cTnT*), cardiac fibroblast (FB) marker (*vimentin*), and endothelial cell (EC) marker (*VE-cad*). (**B**) Cardiac ion channel expression of ZM-treated iPSC-CMs. Results were normalized to GAPDH expression. Data are presented as the mean ± SEM (*n* = 3). * *p* < 0.05, ** *p* < 0.01, by Dunnett’s test.

**Figure 4 biomedicines-09-01952-f004:**
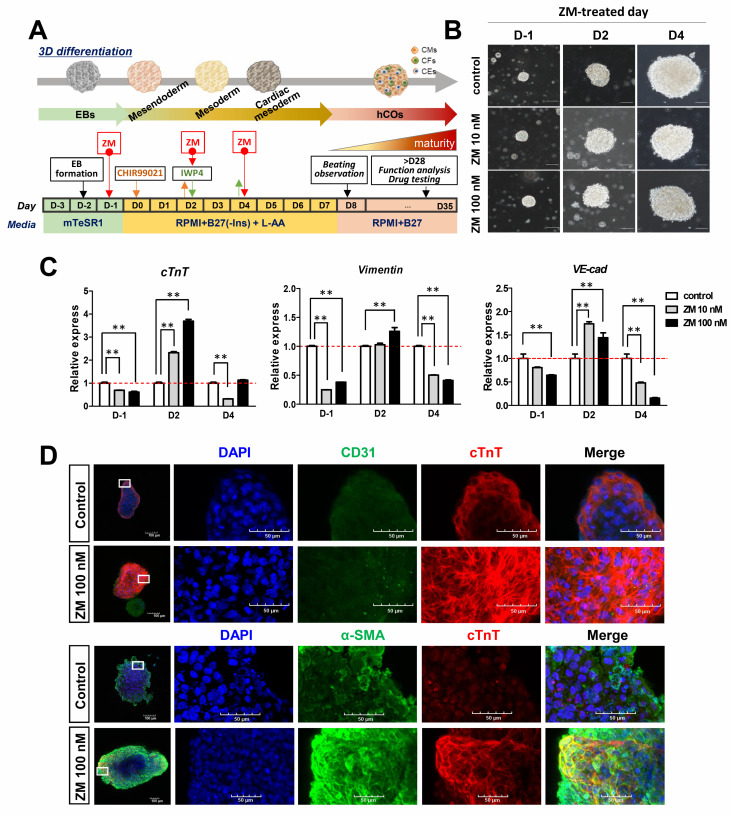
Effects of ZM447439 on the generation of iPSC-hCOs: (**A**) Protocol of hCOs treated with ZM447439. (**B**) Differentiation of 0, 10, or 100 nM ZM treated hCOs. Scale bar, 100 μm. L-AA, L-ascorbic acid. (**C**) Gene expression of mesoderm markers during hCO differentiation normalized to GAPDH expression. Data are presented as the mean + SEM. ** *p* < 0.01, by Dunnett’s test. (**D**) Immunofluorescence to detect CD31 (endothelial marker, green) and cTnT (cardiomyocytes marker, red) (upper). α-SMA (fibroblast marker, green) and cTnT (lower) at iPSC-hCOs.

**Figure 5 biomedicines-09-01952-f005:**
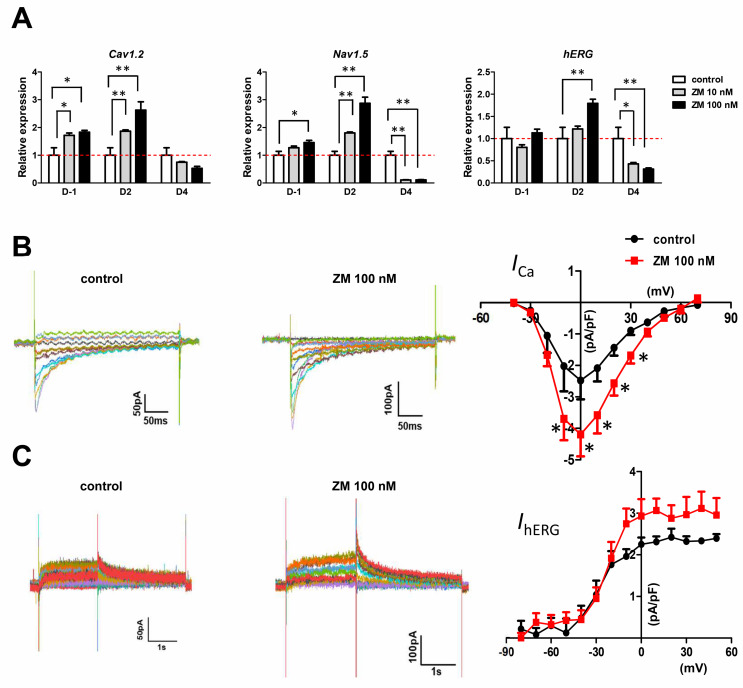
Effects of ZM447439 on cardiac ion channels of iPSC-hCOs: (**A**) Gene expression of cardiac ion channels, *Cav1.2* (calcium channel gene), *Nav1.5* (sodium channel gene), and *hERG* (the human ether-à-go-go-related gene) on iPSC-hCOs. Results were normalized to GAPDH expression. * *p* < 0.05, ** *p* < 0.01 by Dunnett’s test. (**B**) Representative traces demonstrating nifedipine-sensitive currents of *I*_Ca_ in control (left) and ZM-treated (middle) CMs derived from hCOs. I–V relationships of *I*_Ca_ in the control and ZM 100 nM (right) (*n* = 4). (**C**) Representative traces demonstrating E-4031-sensitive rapid component of outward delayed rectifier potassium currents of *I*_hERG_ in control (left) and ZM-treated (middle) cells. I (current)–V (voltage) relationships of *I*_hERG_ in the control and ZM-treated CMs (right) (*n* = 3–5). Data are presented as the mean ± SEM.

**Figure 6 biomedicines-09-01952-f006:**
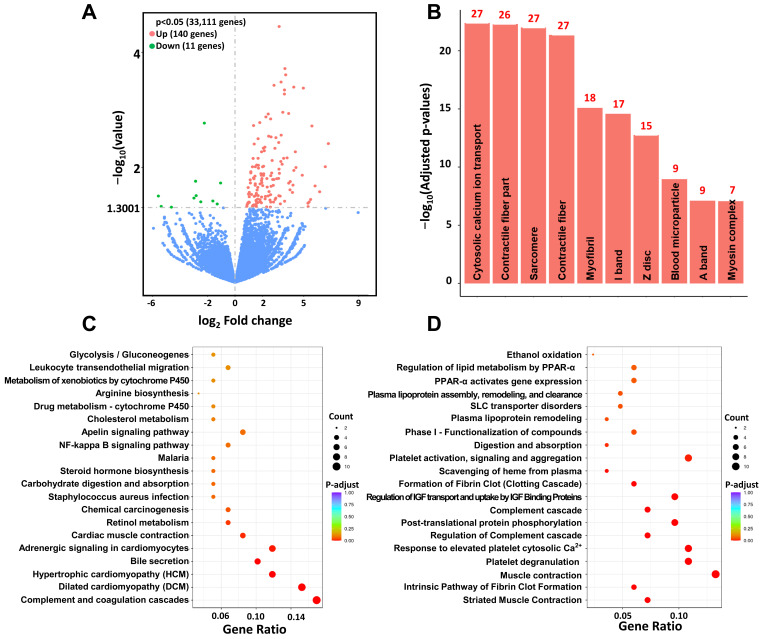
Transcriptome analysis of ZM treated iPSC-hCOs: (**A**) Differential gene volcano map displaying sorted log2 and *p* values. The gray dashed line indicates the threshold line (*p* < 0.05). (**B**) GO cellular components terms enriched in upregulated genes in ZM treated vs. control (adjusted *p* < 0.05). The numbers on the bar graphs represent the number of genes associated with the terms. (**C**) KEGG enrichment analysis, the most significant 20 terms from differentially regulated genes (adjusted *p* < 0.05). (**D**) Most significant 20 Reactome pathways (adjusted *p* < 0.05). PPAR-α, peroxisome proliferator-activated receptor α; IGF, insulin-like growth factor.

**Figure 7 biomedicines-09-01952-f007:**
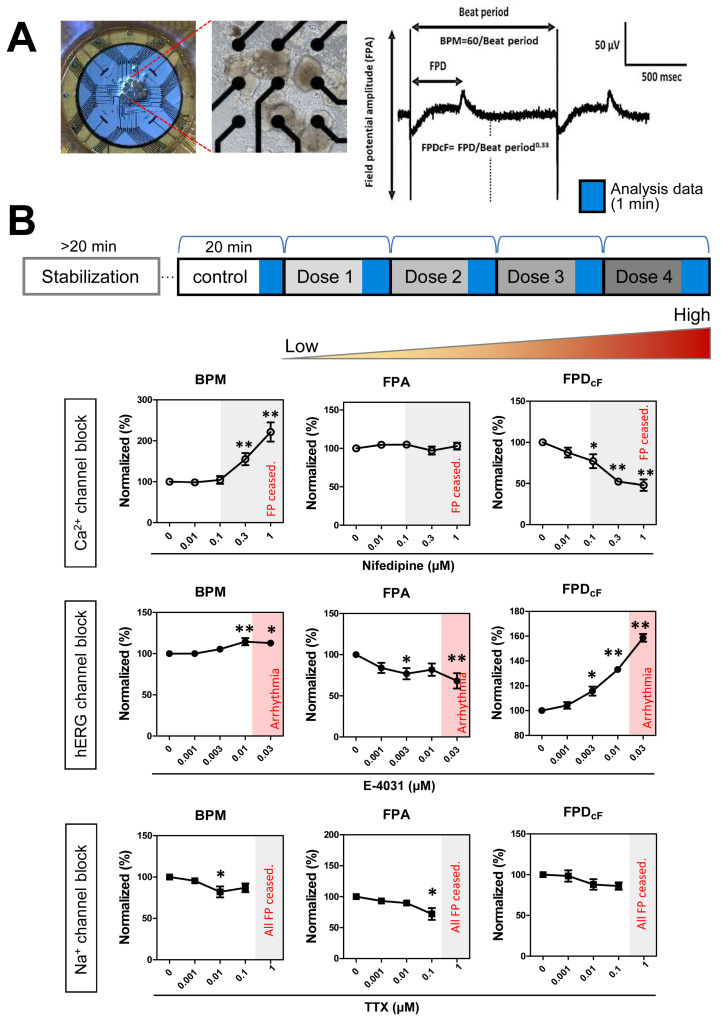
Effects of ion channel blockers on field potentials (FPs) of iPSC-hCOs: (**A**) Photographs of iPSC-hCOs on MEA plate (left) and FP parameters (right). (**B**) Schematic experimental schedules for MEA assay (top) and the effects of nifedipine (*n* = 7), E-4031 (*n* = 6), and Tetrodotoxin (TTX, *n* = 3). BPM, beats per minute; FPA, field potential amplitude; FPD_cF_, Fridericia’s formula to correct the FP duration dependence on beating rate. Data are presented as the mean + SEM. * *p* < 0.05, ** *p* < 0.01, by Dunnett’s test.

**Table 1 biomedicines-09-01952-t001:** List of the primer sequences for qRT-PCR.

Gene Name	Primer Sequence (5′-3′)
*GAPDH*	F: 5′ CAATGGAAATCCCAT-CACCA 3′
	R: 5′ GGGCAGAGATGATGACCCTT 3′
*cTnT*	F: 5′ GGAGAC-CAGGGCAGAAGAAG 3′
	R: 5′ GATCTTGGGAGGCACCAAGT 3′
*Vimentin*	F: 5′ GACAGGATGTTGACAATGCG 3′
	R: 5′ GTTCCTGAATCTGAGCCTGC 3′
*VE-cad*	F: 5′ GAAGCCTCTGATTGGCACAGTG 3′
	R: 5′ TTTTGTGACTCGGAA-GAACTGGC 3′
*Nav1.5*	F: 5′ TTCCTATTACCTCGGGGCAC 3′
	R: 5′ TGCCATAGAGATCTGGCAGC 3′
*Car1.2*	F: 5′ AGTCCAAGACACGGCAAACA 3′
	R: 5′ GCAGTCAAAGCGGTTGAAGA 3′
*hERG*	F: 5′ GAGCAGCCACACATGGACTC 3′
	R: 5′ AGAGCGCCGTCACATACTTG 3′

## Data Availability

The data presented in this study are available on request from the corresponding author.

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
