# Peer review of "Improving Generation of Cardiac Organoids from Human Pluripotent Stem Cells Using the Aurora Kinase Inhibitor ZM447439"

_biomedicines, 2021, doi:10.3390/biomedicines9121952_

Round 1

Reviewer 1 Report

In this study, the authors investigated the effect of ZM447439 on regulating the differentiation of hCOs from hiPSCs. They developed a novel protocol to efficiently and reproducibly generate beating hCOs with improved electrophysiological functionality, contractility, and yield. These findings are interesting. I have several points that need to be specified. Specific points:

  1. In the Methods of 2.3. and 2.4. part, the references should be cited.
  2. There is no information about the primer sequence of GAPDH in the Methods.
  3. Box-plot data from scorecard analysis should be presented in Figure 1B.
  4. Results do not show Figures 2A and 2B.
  5. The results do not match the description in Figure 3A. Furthermore, in Figure 3 it appears that gene levels of VE-cad, Vimentin and hERG are much higher in ZM treatment compared to control, but there is no significant difference.
  6. There is no effect of ZM treatment on the size of hCOs in Figure 4B, however, the size of hCOs in ZM treatment is bigger than control in Figure 4D. Is there any pathophysiologic significance to these findings?
  7. The subtitle of part 3.4 is incorrect.
  8. Has the author compared the effects of ion channel blockers on field potentials of hCOs between control and ZM treatments? These results provide further evidence regarding the advantages of using ZM induced hCOs for drug testing.
  9. The authors need to delve deeper into their findings in Discussion.
  10. The paper needs an extensive editing of punctuation grammatical errors.

Author Response

In this study, the authors investigated the effect of ZM447439 on regulating the differentiation of hCOs from hiPSCs. They developed a novel protocol to efficiently and reproducibly generate beating hCOs with improved electrophysiological functionality, contractility, and yield. These findings are interesting. I have several points that need to be specified. Specific points:

  • Response: We thank you for your careful reading of the manuscript and helpful comments and suggestions. We have made revisions according to your comments and suggestions, as described below. We hope that all the responses could satisfy the reviewer’s comments.
  1. In the Methods of 2.3. and 2.4. part, the references should be cited.
  • Response: According to the reviewer’s comment, we have added references in parts 2.3 and 2.4. In addition, we have modified Fig. 2A and Fig. 4A to clarify the differentiation protocols.
  1. There is no information about the primer sequence of GAPDH in the Methods.
  • Response: To increase readability, we have organized the information for the primer sequences in Table 1. According to the reviewer’s comment, we have added the primer sequence of GAPDH to Table 1 (Page 4, Line 141).
  1. Box-plot data from scorecard analysis should be presented in Figure 1B.
  • Response: As per the reviewer’s comment, we have added the Box-plot data in Figure 1B.
  1. Results do not show Figures 2A and 2B.
  • Response: As pointed out by the reviewer, we forgot to cite Fig. 2A and Fig. 2B in the manuscript. According to the reviewer’s comment, we have cited them in section 3.2.
  1. The results do not match the description in Figure 3A. Furthermore, in Figure 3 it appears that gene levels of VE-cad, Vimentin and hERG are much higher in ZM treatment compared to control, but there is no significant difference.
  • Response: As pointed out by the reviewer, in Fig. 3, the gene expression levels of VE-cad, Vimentin, and hERG are much higher in ZM treatment than in the control group, but there is no significant difference owing to the large variation. We conducted further experiments to clarify this data. The revised figure shows that the gene expression levels of VE-cad, Vimentin, and hERG were significantly higher in ZM treatment than in non-treated controls.
  1. There is no effect of ZM treatment on the size of hCOs in Figure 4B, however, the size of hCOs in ZM treatment is bigger than control in Figure 4D. Is there any pathophysiologic significance to these findings?
  • Response: It’s a very important point. As you know, an aurora kinase inhibitor has an inhibitory effect on cell division, making it large in the case of actively dividing cells. In the previous study, we can confirm the effect with H9C2, stem cells, and cardiomyocytes. Since we use ZM to induce differentiation of stem cells into the heart, it is important to choose non-toxic concentrations, and we selected 10 nM and 100 nM treatment for 24 hours through preliminary experiments.
  • To check the reviewer’s comment, we conducted the size analysis in the ZM treatment groups and the non-treated control group. When comparing the sizes of hCOs in the ZM treatment groups (10 nM or 100 nM for 24 hours) and the non-treated control group, there was no significant difference (below Fig. B). Therefore, the reason why the size of the ZM treatment group is bigger than the control in Fig. 4D is probably due to the difference in individuals between organoids.
  1. The subtitle of part 3.4 is incorrect.
  • Response: We have changed the title to “Increase of cardiac ion channel function in ZM-treated iPSC-hCOs”.
  1. Has the author compared the effects of ion channel blockers on field potentials of hCOs between control and ZM treatments? These results provide further evidence regarding the advantages of using ZM induced hCOs for drug testing.
  • Response: Thank you for your valuable comment. We agree with your advice. Through this study, we focused on demonstrating that the use of an aurora kinase inhibitor, ZM, increases the differentiation efficiency of myocardial cells and organoids, and improves electrophysiological functionality.
  • In this study, we have tested and compared the effect of a calcium channel blocker, nifedipine on field potentials of hCOs between control and ZM treatments. The result shows that the effects of nifedipine did not differ in both groups as follows.
  • As you mentioned, in order to use the ZM induced hCOs for drug testing, the drug reactivity should be checked using many types of drug. We are planning to be carried out as the next project. Thank you.
  1. The authors need to delve deeper into their findings in Discussion.
  • Response: Based on reviewers’ comments, the revision in Discussion includes a number of positive changes.
  1. The paper needs an extensive editing of punctuation grammatical errors.
  • Response: As requested by the reviewer, we have revised both English and grammatical errors throughout the manuscript by Editage (Certificate of language editing use: file:///C:/Users/vanessa/Downloads/Certificate_of_editing-GALEE_4_2_wst8jcczze.pdf).

Reviewer 2 Report

The manuscript ¨ Improving generation of cardiac organoids from human pluripotent stem cells using the aurora kinase inhibitor ZM447439¨ by Lee et al., aims at establishing a new method of hCO differentiation for new drug testing. The authors have performed an extensive and accurate analysis of ZM-treated differentiated hCO, including characterization of their transcriptional profile, functionality and drug reactivity. In the view of this reviewer, the manuscript is of the above-averaged quality, provides interesting data, is written in a direct and very clear way, and can be accepted in the current form for publication in Biomedecines.

I have only some minor comments:

  • Should title 3.4 refer to iPSC-hCOs instead of iPSC-CMs?
  • Figure 5 misses labelling of panel “C”. Moreover, Figure legend 5 refers to I-V, which I do not find on the figure or know what refers to.
  • Figure 5 (concerning section 3.4) refers to E-4031 and nifedipine but explanation of what these drugs are is not detailed until seccion 3.6.  

Author Response

The manuscript ¨ Improving generation of cardiac organoids from human pluripotent stem cells using the aurora kinase inhibitor ZM447439¨ by Lee et al., aims at establishing a new method of hCO differentiation for new drug testing. The authors have performed an extensive and accurate analysis of ZM-treated differentiated hCO, including characterization of their transcriptional profile, functionality and drug reactivity. In the view of this reviewer, the manuscript is of the above-averaged quality, provides interesting data, is written in a direct and very clear way, and can be accepted in the current form for publication in Biomedicines.

I have only some minor comments:

  • Response: Thank you for the careful review and kind comments. We have made revisions according to your comments and suggestions, as described below. We hope that all the responses could satisfy the reviewer’s comments.
  1. Should title 3.4 refer to iPSC-hCOs instead of iPSC-CMs?
  • Response: We have changed the title to “Increase of cardiac ion channel function in ZM-treated iPSC-hCOs.”
  1. Figure 5 misses labeling of panel “C”. Moreover, Figure legend 5 refers to I-V, which I do not find on the figure or know what refers to.
  • Response: We apologize for the typo. Panel “C” has been added to Figure 5. I–V refers to I (current) – V (voltage) relationships. We have added an explanation of I-V on the figure legend.
  1. Figure 5 (concerning section 3.4) refers to E-4031 and nifedipine but explanation of what these drugs are is not detailed until section 3.6.  
  • Response: E-4031 and nifedipine were used as the specific ion channel blockers for the hERG and Ca2+ channels, respectively. We have added an explanation of the drugs in section 3.4. (Page 10, Lines 354-357).

Round 2

Reviewer 1 Report

No further comments